# Effects of Dietary Coated Lysozyme on the Growth Performance, Antioxidant Activity, Immunity and Gut Health of Weaned Piglets

**DOI:** 10.3390/antibiotics11111470

**Published:** 2022-10-25

**Authors:** Xiangfei Xu, Pan Huang, Xuemei Cui, Xuefeng Li, Jiaying Sun, Quanan Ji, Qiang Wei, Yee Huang, Zhefeng Li, Guolian Bao, Yan Liu

**Affiliations:** 1Institute of Animal Husbandry and Veterinary Medicine, Zhejiang Academy of Agricultural Sciences, Hangzhou 310021, China; 2College of Animal Science and Technology·College of Veterinary Medicine, Zhejiang Agricultural and Forestry University, Hangzhou 311300, China; 3Hangzhou King Techina Technology Company Academic Expert Workstation, Hangzhou King Techina Technology Co., Ltd., Hangzhou 311199, China

**Keywords:** antibiotic alternatives, weaned piglets, growth performance, antioxidant activity, immunity, cecal microbiota

## Abstract

The purpose of this study was to evaluate the effects of dietary coated lysozyme on growth performance, serum biochemical indexes, antioxidant activity, digestive enzyme activity, intestinal permeability, and the cecal microbiota in weaned piglets. In total, 144 weaned Large White × Landrace piglets were divided into six treatment groups, with 3 replicates and 8 piglets per replicate: CN, a basal diet; CL-L, CL-M, and CL-H, basal diet supplemented with 100, 150, 500 mg/kg coated lysozyme; UL, basal diet supplemented with 150 mg/kg lysozyme; and Abs, basal diet supplemented with 150 mg/kg guitaromycin for 6 weeks. Compared with the CN and UL diets, dietary CL-H inclusion increased the average daily gain (ADG) and decreased the feed/gain (F/G) ratio of piglets (*p* < 0.05). The addition of 500 mg/kg coated lysozyme to the diet significantly increased the total protein (TP) and globulin (Glob) plasma levels of weaned piglets (*p* < 0.05). Supplementation with 500 mg/kg coated lysozyme significantly increased the serum IgM concentration and increased lipase activity in the duodenum (*p* < 0.05). The addition of coated lysozyme and lysozyme significantly decreased the malondialdehyde (MDA) content, while the superoxide dismutase (SOD), glutathione peroxidase (GSH-Px), and total antioxidant capacity (T-AOC) levels all increased (*p* < 0.05). High-throughput sequencing results showed that CL-H treatment effectively improved the intestinal microbiome. The relative abundance of *Terrisporobacter* in the CL-H and CL-M groups was significantly lower than that in the other groups (*p* < 0.05). LEfSe analysis results showed that the relative abundance of *Coprococcus_3* was higher in the CL-M treatment group. The marker species added to the CL-H treatment group was *Anaerofilum*. In summary, as a potential substitute for feed antibiotics, lysozyme is directly used as a dietary additive, which is inefficient. Therefore, we used palm oil as the main coating material to coat lysozyme. Lysozyme after coating can more effectively improve the growth performance of piglets by improving the intestinal flora, improving the activity of digestive enzymes, reducing the damage to intestinal permeability and oxidative stress in piglets caused by weaning stress, and improving the immunity of piglets.

## 1. Introduction

Weaning is a key stage in the pig breeding cycle. Early weaning often leads to weaning stress in piglets due to changes in food nutrition, the physical environment, psychology and other factors [1,2,3]. Studies have shown that weaning stress can cause changes in the intestinal structure and function of piglets, as well as the intestinal flora, resulting in reduced immunity and oxidative stress. These factors would induce bacterial infections, especially *E. coli* infections, which ultimately lead to diarrhea, decreased growth performance, and, in severe cases, death [4,5,6,7,8]. Weaning stress introduces serious challenges for the pig industry. However, drug resistance has led to a ban on the addition of antibiotics in the diet [9]. Studies have shown that adding antibiotics to the diet disrupts the normal gut flora [10]. Finding a new alternative to antibiotics is particularly important. A large number of studies have confirmed that new feed additives, such as probiotics, enzymes, plant extracts, ZnO, and antibacterial agents, can effectively reduce the use of or replace feed antibiotics [11,12,13]. Lysozyme has bactericidal effects and has the advantages of being natural, green, safe and efficient. Numerous studies have shown that lysozyme can improve the intestinal flora and intestinal morphological structure and is an ideal alternative to antibiotics [14,15,16].

Lysozyme, also known as muramidase or 1,4-β-N-acetylmuramidase, is a natural antibacterial enzyme that is widely detected in saliva, tears and other mammalian secretions [17,18]. It kills bacteria and eliminates pathogens [19,20]. The safety of lysozyme as a dietary supplement in swine was evaluated by Schliffka et al. [21], who found no adverse effects, whereas it improved growth performance. Oliver and Wells [22] showed that lysozyme improved growth performance and gut morphology in finishing pigs. However, many studies have found that the addition of lysozyme to the diet did not improve animal growth performance [16,23]. Studies have shown that heat treatment, high hydrostatic pressure and chemical treatments cause denaturation of lysozyme, thereby reducing lysozyme activity [24,25,26]. Therefore, we expected to improve the effect of lysozyme application by coating lysozyme. At the same time, the coating could improve palatability and facilitate storage. Therefore, this study aimed to investigate the effects of dietary supplementation with coated lysozyme on growth performance, gut digestive enzyme activity, antioxidant activity, gut morphology, immunity, and the cecal bacterial community in weaned piglets.

## 2. Results

### 2.1. Growth Performance

CN, piglets fed a basal diet without any supplementation; CL-L, piglets fed the basal diet supplemented with 100 mg/kg coated lysozyme; CL-M, piglets fed the basal diet supplemented with 150 mg/kg coated lysozyme; CL-H, piglets fed the basal diet supplemented with 500 mg/kg coated lysozyme; UL, piglets fed the basal diet supplemented with 150 mg/kg lysozyme; Abs, piglets fed the basal diet supplemented with 150 mg/kg guitaromycin. The effects of the CL-L, CL-M, CL-H, UL and Abs dietary interventions on growth performance are shown in Table 1. The weaned piglets fed the CL-H diet showed a significantly higher ADG than those fed the basal diet and those fed the UL diet at 14–28 days (*p* < 0.05). During the whole experiment, the ADG values of the CL-H group were significantly higher than those of the CN group and the UL group (*p* < 0.05). The F/G values of the CN group were significantly higher than those of the CL-M, CL-H, UL and Abs groups (*p* < 0.05). At the same time, weaned piglets fed the CL-H diet showed significantly lower F/G values (*p* < 0.05) than those fed the UL diet.

### 2.2. Serum Biochemical Index Analysis

The serum immune indexes of piglets are shown in Figure 1A–F. There were no significant differences in urea indicators (*p* > 0.05) among the three time periods in each group. There were no significant differences in serum Glob values among the groups on day 14 (*p* > 0.05). At 28 days, the Glob value in the serum of the CL-H group was significantly higher than that in the serum of the CN, CL-L and UL groups (*p* < 0.05). The CL-L group exhibited significant differences compared with the CL-H and Abs groups (*p* < 0.05). There was a significant difference in the CN group compared to the CL-L group at 28 days (*p* < 0.05). There was no significant difference in the TP index among the groups at 0 days and 14 days (*p* > 0.05). At 28 days, the CL-H group showed significant improvement compared with the CN, CL-L and UL groups (*p* < 0.05).

### 2.3. Immunoglobulin

The effect of the CL-L, Cl-M, CL-H, UL and Abs dietary interventions on immunoglobulin is shown in Figure 2. The CL-L, Cl-M, and CL-H diets had no significant effect on the IgG content of piglets compared to that of the CN group and UL group on day 28 (*p* > 0.05). The serum IgG content was significantly higher in the CL-H group than in the UL group on day 14 (*p* < 0.01). On day 28, the CN group showed a significantly lower IgM content than the CH-M group (*p* < 0.05) and an extremely significantly lower IgM content than the CL-H and Abs groups (*p* < 0.01), while the CL-H group showed a significantly higher IgM content than the UL group (*p* < 0.05).

### 2.4. Antioxidant Indexes

The effects of the CL-L, CL-M, CL-H, UL and Abs dietary interventions on antioxidant indexes in serum were determined (Table 2). The serum CAT content of the CN treatment group was significantly lower than that of the other groups with dietary supplementation (*p* < 0.05). The CL-L and CL-M treatments significantly increased and decreased the serum GSH-Px content, respectively, compared with other treatments (*p* < 0.05), wherein the UL GSH-Px group had the highest value and the CN group had the lowest value. The weaned piglets fed the basal diet showed a significantly higher MDA index (*p* < 0.05) than the other groups. The CL-L, CL-M and CL-H groups had a significantly lower serum MDA index than the UL group. The SOD index of the serum of the UL group was significantly higher than that in the serum of the other groups (*p* < 0.05), among which that of the CN group was the lowest. The T-AOC indexes in the serum of the CL-H and UL groups were significantly higher than those of the other groups, and the index of the CN group was the lowest (*p* < 0.05).

### 2.5. Digestive Enzymes

The effects of the CN, Cl-L, CL-M, CL-H, UL, and Abs dietary interventions on the activities of digestive enzymes in the duodenum and jejunum are shown in Figure 3A–D. There was no significant difference in the amylase activity in the duodenum and jejunum within all treatment groups (*p* > 0.05). Moreover, the differences in the activities of amylase, lipase, trypsin and chymotrypsin in the jejunum among the groups were not significant (*p* > 0.05). The lipase activity in the duodenum in the CL-H group was significantly higher than that in the Abs group (*p* < 0.05). The trypsin activity in the duodenum in the Abs group was significantly higher than that in the CL-L group (*p* < 0.05).

### 2.6. Morphological Analysis of the Small Intestine

The effects of the CL-L, CL-M, CL-H, UL and Abs dietary interventions on the morphology of the small intestinal mucosa is shown in Figure 4A–C. No significant difference (*p* > 0.05) was found in the intestinal wall thickness or V/C value of the small intestine (duodenum, jejunum, ileum) within any of the treatment groups. HE staining microscopy showed that the intestinal villus structure of CL-H-treated piglets was more intact and denser than that of the control and UL-fed piglets. Intestinal villi were dissolved in both the CN group and UL group.

### 2.7. Quantification of Short-Chain Fatty Acids

The effects of the CL-L, CL-M, CL-H, UL and Abs dietary interventions on short-chain fatty acid levels in the colonic contents are shown in Table 3, including on the levels of acetic acid, propionic acid, isobutyric acid, butyric acid, isovaleric acid, and valeric acid. There was no significant difference in the isobutyric acid or isovaleric acid level in the colonic contents within all treatment groups. The valeric acid concentration of the colonic contents in the CL-H group and Abs group was significantly higher than that in the other groups.

### 2.8. 16S rRNA Sequencing of the Cecal Microflora

The 16S rRNA sequencing results of the cecal contents are shown in Figure 5A–D and Figure 6A,B. According to the alpha diversity analysis, the Shannon indexes was not significantly different among the groups. There were 529 core OTUs observed in the Venn diagram of the six groups. 38, 23, 47, 44, 34 and 19 elements were unique to the CN, CL-L, CL-M, CL-H, UL and Abs groups, respectively. The relative abundance of *Terrisporobacter* in the CL-H and CL-M groups was significantly lower than that in the other groups (*p* < 0.05). The relative abundance of *Lachnospiraceae_AC2044_group* in the CL-L group was significantly higher than that in the other groups (*p* < 0.05). Beta diversity analysis and comparison between samples, calculation of the weighted UniFrac distance index, and Welch’s t test were also applied. The CN group index was significantly lower than that of the other groups (*p* < 0.05). PCA showed that the gut microbial composition of the UL group piglets was significantly different from that of the other groups. To further investigate changes in the fecal bacterial composition, we employed LEfSe analysis to identify the most differentially abundant genera across all treatment groups. To further study the changes in the cecal microbial composition, LEfSe analysis was performed to screen the genera with significant differences in abundance among the treatment groups. The CL-L dietary intervention increased the relative abundance of *Terrisporobacter* and *Peptostreptoccaceae* in the cecal microbiota. The relative abundance of *Coprococcus_3* and *Sphaerochaeta* was higher in the CL-M treatment group. The marker species added to the CL-H treatment group were *Anaerofilum* and *Lachnospiraceae*. PICRUSt analysis was performed to examine the KEGG pathways of the cecal microbiota. Second-level KEGG analysis showed that carbohydrate metabolism, amino acid metabolism, biosynthesis of other secondary metabolites, membrane transport, and environmental adaptation were enriched in CL-H-treated piglets compared with the observations for the control piglets. Meanwhile, metabolism of terpenoids and polyketides and biosynthesis of other secondary metabolites were enriched in the CL-M-fed piglets compared with the observations for the UL-fed piglets.

## 3. Discussion

The addition of lysozyme to the diet has multiple benefits for animals. Previous studies have shown that adding lysozyme to animal diets can improve growth performance and immunity, reduce the diarrhea rate, improve the small intestinal morphology, etc. [27,28]. However, experiments by Humphrey BD and Xia Y [16,23] showed that lysozyme did not enhance the growth performance of experimental animals. Thus, we expected to improve the economic benefits of using lysozyme as a feed additive by coating it. The data from this study showed that the ADG of the CL-H treatment group was significantly higher than that of the CN and UL groups, and the F/G value of the CL-H treatment group was significantly lower than that of the CN and UL groups. The lower feed intake in the UL group may be due to its poor palatability. It was concluded that lysozyme can improve the growth performance of piglets after coating compared with uncoated lysozyme.

The bones, muscles, and intestines of pigs grow densely, and many nutrient metabolism processes change after weaning, so their serum biochemical parameters are constantly changing [29,30]. Serum biochemical indicators directly show the growth performance of animals by reflecting the immune function and protein metabolism level of the body. Metabolic abnormalities caused by low protein content in the diet are reflected by the serum parameters. TP includes albumin and globulin. Albumin is mainly a protein that grows in the liver and is related to metabolism, and globulin is mainly an immunoglobulin [31,32,33]. The main aspect of muscle growth is protein deposition, which is closely related to protein synthesis and metabolism [34]. In this study, the addition of 500 mg/kg coated lysozyme to the diet significantly increased the TP and Glob levels of weaned piglets. This finding shows that coated lysozyme can promote the synthesis and metabolism of proteins and promote the growth of tissues and organs. IgG and IgM are immunoglobulins in serum, and they are important effectors of humoral immunity. IgG is the main antibody involved in the secondary immune response, and IgM is the earliest produced immunoglobulin in humoral immunity. In this study, dietary supplementation with 500 mg/kg coated lysozyme increased only the IgG concentration at 14 days, while dietary supplementation with 500 mg/kg coated lysozyme significantly increased the serum IgM concentration, which is consistent with the findings of Xu et al. [28], who reported similar results with the addition of lysozyme. During this experiment, none of the experimental piglets became sick, indicating that adding coated lysozyme to the diet can promote the immune index of weaned piglets.

Antioxidative enzymes protect the body from oxidative damage while enhancing the body’s defense function, so intestinal antioxidants are extremely important for bodily health [35]. MDA is the final product of lipid peroxidation in the body, which can damage cell structure and function, and its content can indirectly reflect the degree of damage to the body [36]. SOD, GSH-Px, CAT, and T-AOC can remove excess free radicals in the body, maintain the homeostasis of the internal environment, and prevent the damage caused by free radicals to the body. Therefore, the higher the content of these indicators is, the stronger the body’s antioxidant capacity [37,38]. Studies have shown that weaning stress can lead to an increase in the MDA content in the serum of piglets and a decrease in the levels of SOD, GSH-Px, CAT, T-AOC, etc. [39,40,41]. In this study, the addition of coated lysozyme and lysozyme significantly decreased the MDA content while increasing the SOD, GSH-Px, and T-AOC levels. This result showed that both lysozyme and coated lysozyme help improve the body’s antioxidant capacity. However, the specific mechanism remains to be studied.

The activity of digestive enzymes directly affects the digestibility of feed and the growth performance of animals. The breakdown of protein, starch and cellulose requires the synergistic action of different digestive enzymes, and any changes in the activity of digestive enzymes affect this process [42,43,44]. When piglets are weaned, the pH value in the gastrointestinal tract often increases due to insufficient lactose, and the activity of digestive enzymes is reduced, which leads to the growth of harmful intestinal bacteria and endangers the intestinal health of piglets [45]. Therefore, enhancing the activity of digestive enzymes in weaned piglets urgently needs to be addressed. In the present study, dietary supplementation with 500 mg/kg coated lysozyme was found to increase lipase activity in the duodenum, thus indicating that the addition of coated lysozyme to the diet could improve the digestive enzyme activity in the gastrointestinal tract of weaned piglets.

According to our results, dietary supplementation with 500 mg/kg coated lysozyme significantly increased the level of valeric acid in the colonic contents of weaned piglets. Short-chain fatty acids are metabolites of intestinal anaerobic microorganisms that can not only provide energy for intestinal epithelial cells but also play a role in regulating the intestinal flora [46,47]. In addition, valeric acid has the effect of inhibiting the growth of breast cancer cells and protecting dopaminergic neurons [48,49]. A large number of intestinal bacteria form a complex microbial system with various functions, such as digestion, absorption, metabolism, and immunity [50,51]. Studies have shown that when the body is invaded by external pathogens, the gut microbiota can stimulate the body to produce antibacterial compounds to defend against the pathogens [52]. In the present study, Venn diagram analysis showed that all the samples of the six groups shared a large microbiome, but each group also had its own unique microbiome. Moreover, *Terrisporobacter* has also been shown to induce oxidative stress [53]. In this study, the addition of 150 mg/kg coated lysozyme to the diet significantly reduced the intestinal *Terrisporobacter* abundance. This result is consistent with the result that the addition of coated lysozyme significantly reduces the MDA content and further proves that coated lysozyme can enhance the antioxidant activity in the body. The LEfSe analysis of marker species showed the differentially abundant species after supplementation with 150 mg/kg coated lysozyme in the diet. The specific species *Coprococcus_3* in the CL-H group can regulate intestinal permeability and scavenge free radicals [54]. The specific species *Anaerofilum* in the CL-H group has been shown to be positively correlated with the body weight of the animals [55]. In conclusion, coated lysozyme can effectively improve the intestinal microbiome.

## 4. Materials and Methods

### 4.1. Animals, Treatment, and Designation

Piglets were obtained from the experimental pig farm of the Animal Husbandry and Veterinary Research Institute, Zhejiang Academy of Agricultural Sciences (Hangzhou, China). All feed additives used in the experiments were provided by Zhejiang Kangdequan Technology Co., Ltd. (Hangzhou, China). Lysozyme was sprayed with palm oil as a coating material to form granules. The lysozyme used in this experiment was a lysozyme base with an activity of 2 × 10^7^ U/g. The lysozyme activity after coating was 5 × 10^6^ U/g. The pig house was 5.5 m long and 5 m wide, with 18 pigs in total. The room temperature was maintained at 22–25 °C, and the humidity was 60–70%. All the piglets had free access to food and water throughout the 28-day feeding trial.

### 4.2. Experimental Design

This animal experiment was approved by the Ethics Committee of Zhejiang Academy of Agricultural Sciences. Animal studies were conducted in accordance with the principles and guidelines of the Zhejiang Provincial Farm Animal Welfare Committee of China. A total of 144 weaned Large White × Landrace piglets (8.66 ± 0.33 kg, half male and half female) were selected according to age and weight and divided into 6 treatment groups, with 3 replicates and 8 piglets per replicate: (1) the control group (CN; fed a basal diet without any supplementation); (2) the low-dose coated lysozyme group (CL-L; fed the basal diet supplemented with 100 mg/kg coated lysozyme); (3) the medium-dose coated lysozyme group (CL-M; fed the basal diet supplemented with 150 mg/kg coated lysozyme); (4) the high-dose coated lysozyme group (CL-H; fed the basal diet supplemented with 500 mg/kg coated lysozyme); (5) the uncoated lysozyme group (UL; fed the basal diet supplemented with 150 mg/kg lysozyme); and (6) the antibiotic group (Abs; fed the basal diet supplemented with 150 mg/kg guitaromycin). The basal feed was formulated as shown in Table 4.

### 4.3. Sample Collection

Twenty-eight days later, a total of 36 (2 per replicate) healthy piglets were randomly selected from each experimental group and killed by euthanasia with an intravenous (i.v.) injection of sodium pentobarbital (40 mg/kg body weight). Blood samples were collected via the anterior vena cava on days 1, 14, 21 and 28. On day 28, three 5 cm-long segments were acquired from the middle duodenum, jejunum and ileum. One 5 cm-long fresh tissue sample (duodenum, jejunum, and ileum) was washed with normal saline and fixed in 4% paraformaldehyde for villus morphology assessment. The remaining two 5-cm-long intestinal segments were immediately stored at −80 °C until analysis.

### 4.4. Growth Performance Evaluation

Growth performance was evaluated by body weight, average daily gain (ADG), and feed-to-weight ratio (F/G). We weighed the piglets on days 1, 14 and 28, recorded the feed intake, and calculated the ADG and F/G.

### 4.5. Determination of Serum Biochemical Index and Immunoglobulin Levels

Blood samples were collected from the piglets via the anterior vena cava in a 5 mL tube on days 14 and 28. Blood samples were centrifuged (3000× *g*, 10 min) at 4 °C for serum collection, and the obtained serum was immediately stored in Eppendorf tubes at −20 °C until further analysis. Biochemical parameters, including total cholesterol (TChol), albumin (Alb), alkaline phosphatase (AL), globulin (Glob), total protein (TP), and urea plasma levels and the albumin-globulin ratio (A/G), were determined using a GS200 Automatic Biochemical Analyzer (Hangzhou Genius Electronics Co., Ltd., Hangzhou, China). Immunoglobulins were quantified using an ELISA kit purchased from Shanghai Enzyme Link Biotechnology Co., Ltd (Shanghai, China). The procedure was performed according to the instructions of the manufacturer.

### 4.6. Determination of Antioxidant Indexes

The antioxidant indexes of the day 28 sample, including the glutathione peroxidase (GSH-Px) and superoxide dismutase (SOD) activities, catalase (CAT) plasma levels, total antioxidant capacity (T-AOC) and malondialdehyde (MDA) plasma content, were measured using reagent kits purchased from Nanjing Jian Cheng Bioengineering Institute (Nanjing, China) following the manufacturer’s instructions.

### 4.7. Digestive Enzymes

Amylase, lipase, trypsin, and chymotrypsin were detected with a BCA test kit from Nanjing Jiancheng Bioengineering Institute (Nanjing, China). The procedure was performed according to the instructions of the manufacturer.

### 4.8. Morphological Analysis of the Small Intestine

The small-intestinal tissue was completely immersed in 4% paraformaldehyde solution for at least 24 h, dehydrated, dipped in wax, embedded, sectioned, stained, and sealed. An Eclipse Ci-L photographic microscope was used to select the target area of the tissue for 40× imaging, and the background light of each photo was kept consistent during imaging. Image-Pro Plus 6.0 analysis software was used to measure the length of 5 intact villi (V) and the depth of 5 crypts (C) in each slice, and the average value was calculated. Then, the V/C ratio was calculated.

### 4.9. Determination of Short-Chain Fatty Acid Levels

First, 1 mL of distilled water was added to 0.5 g of colonic contents. After mixing, the supernatant was obtained by centrifugation, a mixed solution of metaphosphoric acid was added, and the mixture was frozen at −20 °C for more than 24 h. After thawing, the samples were centrifuged and filtered through a 0.22 μm filter. The concentration of short-chain fatty acids in the samples was determined using a GC-2010 Plus gas chromatograph (Shimadzu, Kyoto, Japan).

### 4.10. 16S rRNA Sequencing of the Cecal Microflora

Microbial DNA was extracted using HiPure Stool DNA Kits (Magen, Guangzhou, China) according to the manufacturer’s protocols. The V3–V4 region of the standard bacterial 16S rRNA gene was subsequently amplified with specific primers (F: 5′-CCTACGGGNGGCWGCAG-3′ and R: 5′-GGACTACHVGGGTATCTAAT-3′). Read filtering of sequences was performed using FASTP, and the sequences were subjected to quality optimization and filtered using FLASH and UPARSE software. The representative OTU sequences or ASV sequences were classified into organisms by a naive Bayesian model using the RDP classifier (version 2.2) based on the SILVA (version 132), UNITE (version 8.0) or ITS2 database (version update_2015), with a confidence threshold value of 0.8. The abundance statistics of each taxon were visualized using Krona (version 2.6). Between-group Venn analysis was performed in the R package VennDiagram (version 1.6.16). Shannon indexes were calculated in QIIME (version 1.9.1). Tukey’s honestly significant difference (HSD) test was calculated in the R package Vegan. Through multiple KEGG functional cluster abundance analyses, the changes in functions in different treatment groups were analyzed.

### 4.11. Statistical Analysis

All experimental data were analyzed using the SPSS 19.0 statistical software package (SPSS Inc., Chicago, IL, USA) and GraphPad Prism 8.0 software package (GraphPad Software Inc., La Jolla, CA, USA). The experimental data were analyzed using one-way ANOVA and Duncan’s multiple comparison test. All the data are expressed as the mean ± standard deviation (M ± SE), and *p* values less than 0.05 were considered significant.

## 5. Conclusions

This study showed that dietary supplementation with coated lysozyme could improve the growth performance, antioxidant capacity, and the gut microbial community and diversity of weaned piglets. Among the interventions, the addition of high-dose coated lysozyme (500 mg/kg) can improve the immunity and intestinal digestive enzyme activity of weaned piglets and increase the valeric acid levels in the cecum.

## Figures and Tables

**Figure 1 antibiotics-11-01470-f001:**
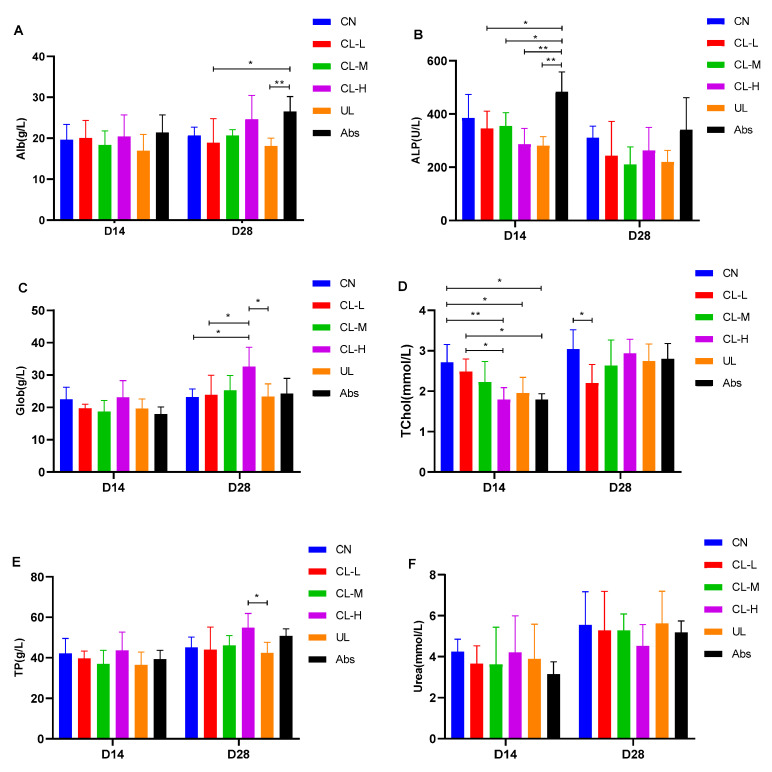
Effects of the CN, CL-L, CL-M, CL-H, UL, and Abs dietary interventions on the serum immunologic indexes of piglets. (**A**–**F**) Albumin (Alb), alkaline phosphatase (ALP), globulin (Glob), total cholesterol (TChol), total protein (TP), and urea levels and albumin–globulin ratio (A/G). The data are presented as the means ± SE (*n* = 6). * indicates significant differences (*p* < 0.05), ** represents very significant differences (*p* < 0.01).

**Figure 2 antibiotics-11-01470-f002:**
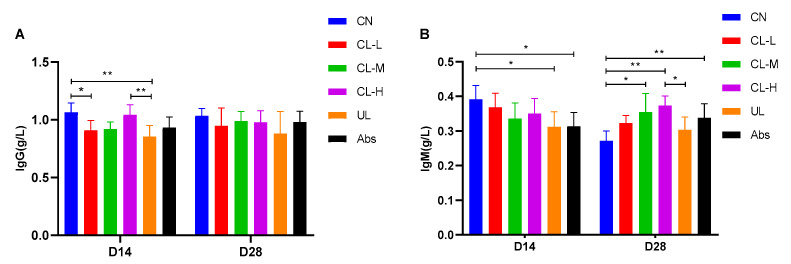
Effects of the CL-L, CL-M, CL-H, UL and Abs dietary interventions on serum immunoglobulins. (**A**,**B**) represent IgG and IgM, respectively. The data are presented as the means ± SE (*n* = 6). * indicates significant differences (*p* < 0.05), ** represents very significant differences (*p* < 0.01).

**Figure 3 antibiotics-11-01470-f003:**
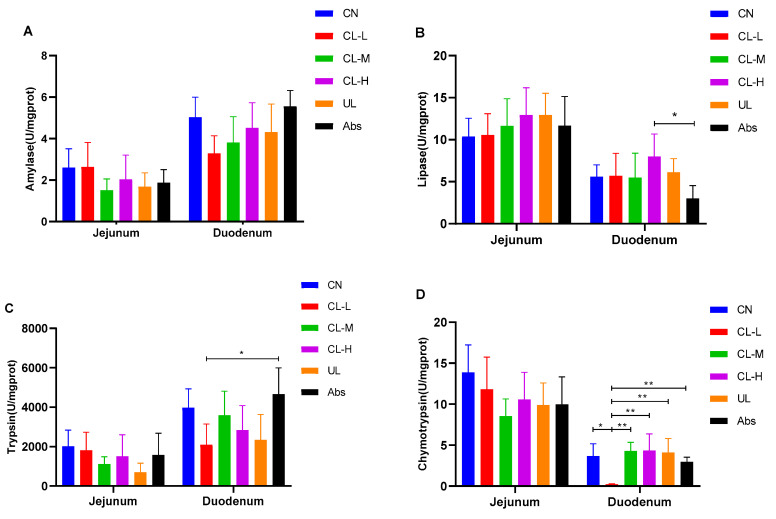
Effects of the CN, CL-L, CL-M, CL-H, UL, and Abs dietary interventions on (**A**–**D**) amylase, lipase, trypsin and chymotrypsin activities in the duodenum and jejunum. The data are presented as the means ± SE (*n* = 6). * indicates significant differences (*p* < 0.05), ** represents very significant differences (*p* < 0.01).

**Figure 4 antibiotics-11-01470-f004:**
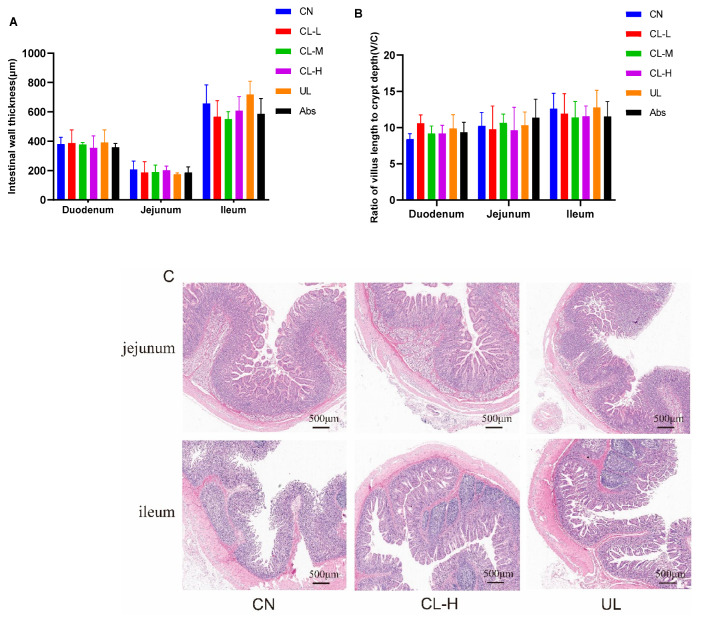
Effects of the CL-L, CL-M, CL-H, UL and Abs dietary interventions on small intestinal mucosal morphology. (**A**–**C**) intestinal wall thickness, ratio of villus length to crypt depth (V/C) and HE staining microscopy results (40×). The data are presented as the means ± SE (*n* = 4).

**Figure 5 antibiotics-11-01470-f005:**
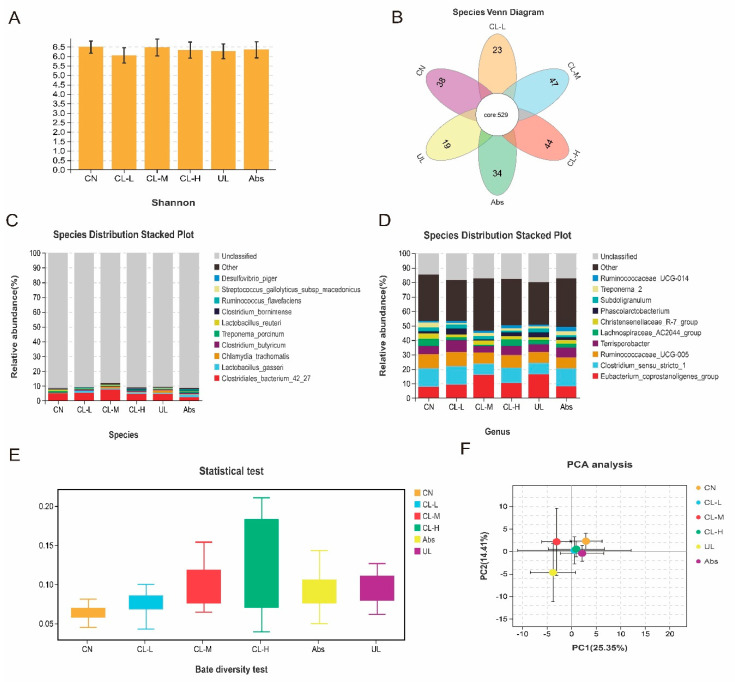
Summary of the microbial community in the cecal contents of piglets. (**A**) Shannon index in the alpha diversity analysis. (**B**) Venn diagram of OTUs. (**C**) Species-level species distribution stacked plot. (**D**) Genus-level species distribution stacked plot. (**E**) Statistical tests for beta diversity analysis. (**F**) PCA.

**Figure 6 antibiotics-11-01470-f006:**
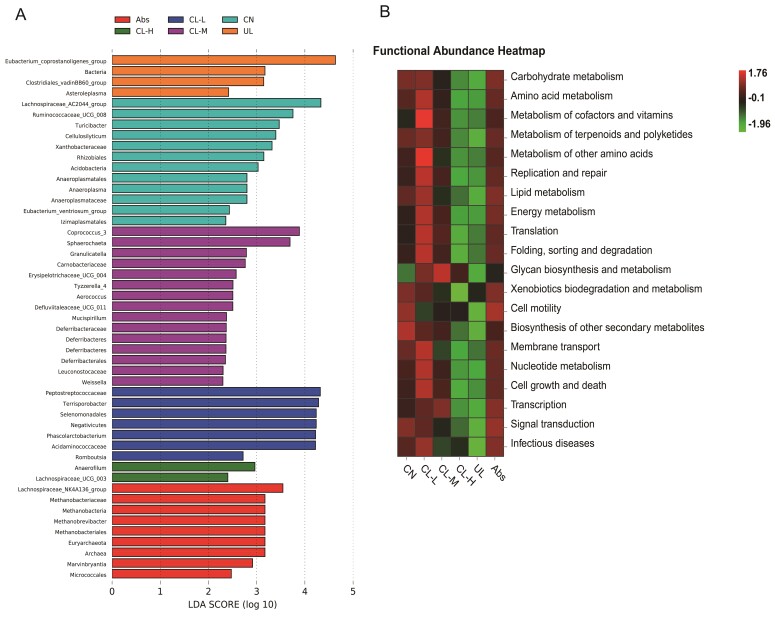
Summary of the microbial community in the cecal contents of piglets. (**A**) LEfSe. (**B**) Second-level KEGG pathway analysis of the predicted functions of the colonic microbiota.

**Table 1 antibiotics-11-01470-t001:** Effects of the CN, CL-L, CL-M, CL-H, UL, and Abs dietary interventions on the growth performance of piglets.

	CN	CL-L	CL-M	CL-H	UL	Abs	SEM	*p* Value
(d 0) Initial weight (kg)	8.66 ± 0.95	8.71 ± 0.31	9.18 ± 0.24	8.91 ± 0.25	8.62 ± 0.34	9.08 ± 0.49	85.07	0.299
(d 14) Middle weight (kg)	12.74 ± 0.80 ^c^	13.11 ± 0.25 ^c^	14.26 ± 0.51 ^abc^	14.60 ± 0.31 ^ab^	13.78 ± 1.12 ^bc^	16.06 ± 0.19 ^a^	222.54	0.007
(d 28) Final weight (kg)	16.88 ± 1.30 ^b^	1.81 ± 0.19 ^b^	19.07 ± 0.77 ^b^	20.84 ± 0.67 ^a^	18.47 ± 1.60 ^b^	21.98 ± 0.47 ^a^	373.22	0.000
ADG (g)								
Days 0 to 14	339.13 ± 49.85 ^bc^	313.87 ± 14.54 ^c^	362.67 ± 35.29 ^bc^	406.24 ± 4.50 ^ab^	368.33 ± 61.36 ^bc^	451.23 ± 21.60 ^a^	13.08	0.009
Days 14 to 28	295.71 ± 40.92 ^c^	358.77 ± 22.72 ^abc^	343.62 ± 18.86 ^bc^	445.74 ± 68.57 ^a^	334.91 ± 81.18 ^bc^	422.80 ± 41.55 ^ab^	16.09	0.029
Days 0 to 28	317.42 ± 24.74 ^b^	336.32 ± 8.75 ^b^	353.14 ± 25.86 ^b^	425.99 ± 32.48 ^a^	351.62 ± 55.86 ^b^	437.01 ± 26.31 ^a^	12.64	0.003
ADFI (g)	672.57 ± 4.92 ^a^	613.50 ± 11.54 ^b^	605.60 ± 10.66 ^b^	649.26 ± 22.38 ^a^	544.97 ± 12.35 ^c^	673.90 ± 21.50 ^a^	11.31	0.000
F/G	2.13 ± 0.15 ^a^	1.82 ± 0.05 ^b^	1.72 ± 0.11 ^b^	1.53 ± 0.06 ^c^	1.58 ± 0.28 ^b^	1.54 ± 0.05 ^c^	0.06	0.002

^a,b^^,c^ mean values within a row with unlike superscript letters were significantly different (*p* < 0.05) ADG, average daily gain; ADFI, average daily feed intake; F/G, feed-to-weight ratio. The data are presented as the means ± SE (n = 24).

**Table 2 antibiotics-11-01470-t002:** Antioxidant indicators in serum.

	CN	CL-L	CL-M	CL-H	UL	Abs	SEM	*p* Value
CAT (U/mL)	169.01 ± 56.16 ^b^	242.58 ± 51.59 ^a^	263.70 ± 50.18 ^a^	281.09 ± 44.31 ^a^	280.33 ± 54.93 ^a^	291.10 ± 50.60 ^a^	10.77	0.003
GSH-Px (U/L)	80.03 ± 11.93 ^d^	105.68 ± 23.53 ^c^	103.72 ± 13.89 ^c^	135.71 ± 18.13 ^ab^	145.50 ± 14.28 ^a^	118.44 ± 20.24 ^bc^	4.62	0.000
MDA (nmol/mL)	5.97 ± 1.20 ^a^	5.61 ± 1.12 ^ab^	4.20 ± 1.04 ^bc^	4.39 ± 1.38 ^bc^	3.25 ± 1.37 ^c^	5.39 ± 1.04 ^ab^	0.25	0.005
SOD (U/mL)	195.03 ± 45.01 ^d^	258.78 ± 39.32 ^ab^	300.52 ± 51.47 ^abc^	329.88 ± 52.14 ^ab^	343.96 ± 53.61 ^a^	270.14 ± 63.65 ^bc^	11.79	0.000
T-AOC (μmol/mL)	1.54 ± 0.31 ^c^	2.05 ± 0.23 ^b^	2.19 ± 0.35 ^ab^	2.49 ± 0.54 ^a^	2.58 ± 0.25 ^a^	2.07 ± 0.35 ^b^	0.08	0.000

^a,b,c^^,d^ mean values within a row with unlike superscript letters were significantly different (*p* < 0.05) The data are presented as the means for the antioxidant index; MDA, malondialdehyde; SOD, superoxide dismutase, T-AOC, total antioxidant capacity. The data are presented as the means ± SE (n = 6).

**Table 3 antibiotics-11-01470-t003:** Determination of short-chain fatty acid concentrations (mM).

	CN	CL-L	CL-M	CL-H	UL	Abs	SEM	*p* Value
acetic acid	22.54 ± 8.77 ^a^	14.64 ± 2.30 ^c^	13.42 ± 2.90 ^c^	15.74 ± 2.41 ^ab^	14.47 ± 2.75 ^c^	17.53 ± 4.72 ^ab^	1.04	0.113
propionic acid	8.71 ± 2.85 ^a^	6.34 ± 0.66 ^ab^	5.40 ± 1.34 ^b^	6.39 ± 0.76 ^ab^	5.87 ± 0.58 ^b^	7.80 ± 2.42 ^ab^	0.39	0.100
isobutyric acid	0.66 ± 0.15 ^a^	0.56 ± 0.09 ^a^	0.44 ± 0.12 ^a^	0.61 ± 0.14 ^a^	0.60 ± 0.16 ^a^	0.63 ± 0.29 ^a^	0.03	0.521
butyric acid	4.16 ± 1.05 ^a^	2.60 ± 0.61 ^bc^	1.92 ± 0.74 ^c^	2.69 ± 0.43 ^bc^	2.13 ± 1.07 ^bc^	3.37 ± 0.62 ^ab^	0.21	0.008
isovaleric acid	1.11 ± 0.25 ^a^	0.90 ± 0.16 ^a^	0.72 ± 0.18 ^a^	1.05 ± 0.20 ^a^	1.05 ± 0.35 ^a^	1.15 ± 0.41 ^a^	0.06	0.284
valeric acid	0.80 ± 0.16 ^ab^	0.68 ± 0.08 ^ab^	0.59 ± 0.10 ^b^	0.85 ± 0.09 ^a^	0.71 ± 0.14 ^ab^	0.87 ± 0.21 ^a^	0.03	0.075

^a,b,c^^,^ mean values within a row with unlike superscript letters were significantly different (*p* < 0.05) The data are presented as the means ± SE (*n* = 4).

**Table 4 antibiotics-11-01470-t004:** Composition and nutrient levels of the basal diet.

Diet Composition	Percentage (%)	Nutrient Levels	Content
Corn	62.12	Total (MJ/kg)	14.55
Bean meal	24.88	Crude protein	17.4
Wheat bran	3	Total phosphorus	0.51
Fatty powder	2.0	Lysine	1.158
Limestone	1.2	Methionine+cysteine	0.612
Calcium hydrogen phosphate	0.8	Threonine	0.679
Soybean oil	0.07		
Fish meal	1		
NaCl	0.3		
Choline chloride	0.1		
Potassium magnesium sulfate	0.2		
Premix *	1		

* Provided per kilogram of diet: 16,000 IU of vitamin A, 4000 IU of vitamin D3, 100 IU of vitamin E, 0.5 mg of vitamin K3, 2 mg of vitamin B1, 4.5 mg of vitamin B2, 7 mg of vitamin B6, 0.03 mg of vitamin B12, 0.2 mg of biotin, 10 mg of folic acid, 30 mg of nicotinic acid, 22 mg of pantothenic acid, 85 mg of Fe (FeSO_4_), 100 mg of Cu (CuSO_4_), 0.3 mg of Mn (MnSO_4_), and 0.14 of mg I (CaI_2_).

## Data Availability

Data are contained within the article; there is no supplementary material.

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
