# Peer review of "Effects of Dietary Coated Lysozyme on the Growth Performance, Antioxidant Activity, Immunity and Gut Health of Weaned Piglets"

_antibiotics, 2022, doi:10.3390/antibiotics11111470_

Round 1
Author Response
Dear Reviewer,
Thank you very much for reviewing our manuscript (Antibiotics-1946742) and for your valuable comments and suggestions. They are very helpful for improving our manuscript. According to your comments and suggestions, we thoroughly revised our manuscript and submitted it to Antibiotics for further review. The changes based on your suggestions were highlighted in red colour in the new version. Point by point corrections were made as follows:
Integration to the Abstract is required: the article describes the effects of dietary enveloped lysozyme on growth performance, serum biochemical indexes, antioxidant activity, digestive enzyme activity, intestinal permeability, and the caecal microbiota in weaned piglets. The term “enveloped lysozyme” is a generic definition that the Authors specify only in the section “4. Materials and methods (lines 337-338)”. A sentence should be added in the Abstract to clarify what they mean for “enveloped lysozyme” since this is the real subject matter of the study.
Response 1: Thank you very much for your comments. “We used palm oil as the main coating material to coat lysozyme”. Please see line 35.
In the section “4. Materials and Methods” the Authors should specify if the term lysozyme they use refers to Lysozyme base or to Lysozyme Hydrochloride or citrate and the purity of the same.
Response 2: Thank you. I have added lysozyme as lysozyme base in the manuscript and marked its concentration as 2×107. Please see lines 307-308.
In the section “4. Material and methods” line 338-specify the enzymatic test utilized for the determination of the enzymatic activity (500 U/g) on the “enveloped lysozyme”. It is the traditional Micrococcus luteus test? And which is the enzymatic activity of the native Lysozyme utilized for the preparation of the “enveloped lysozyme”?
Response 3: Thank you. The method of detecting enzyme activity is Micrococcus luteus test. The lysozyme activity of the raw material used to prepare the enveloped lysozyme is 2×107U/g. Please see lines 307-308.
Clarify the rational of the comparative test with guitataromycin (Abs)
Response 4: Thank you. A large number of literatures show that guitarmycin as a feed additive can improve animal growth performance and prevent bacterial infection. It is included in the background of the world’s anti-antibiotic prohibition, and at the same time, guitaromycin is also one of the most commonly used antibiotic additives. I list 2 similar literatures using guitarmycin as a positive control.
Ma J, Piao X, Shang Q, Long S, Liu S, Mahfuz S. Mixed organic acids as an alternative to antibiotics improve serum biochemical parameters and intestinal health of weaned piglets. Anim Nutr. 2021 Sep;7(3):737-749.
Ma J, Shang Q, Long S, Liu S, Mahfuz SU, Piao X. PSVII-10 Growth performance, serum biochemical parameters and intestinal health of piglets as affected by dietary moa at different levels. J Anim Sci. 2021 Oct 8;99(Suppl 3):410.
Separate the references in brackets with a space from the text (this for all the cited references in the text)
Response 5: Thank you. I have put spaces before all brackets.
Lines 62-63. Correct the phrase from “Experiments conducted by Ferraboschi et al.[22] illustrated the feasibility of antibiotic replacement by lysozyme” to “A recent review published by Ferraboschi et al.[22] illustrated the feasibility of antibiotic replacement by lysozyme”or a similar phrase. In fact, the cited reference is a review and does not contain an experimental section.
Response 6: Thank you. According to your valuable comments,“Experiments conducted by Ferraboschi et al.[22] illustrated the feasibility of antibiotic replacement by lysozyme” was changed into “A recent review published by Ferraboschi et al.[22] illustrated the feasibility of antibiotic replacement by lysozyme”. Please see lines 61-63.
Line 77. Correct “Table 2” with “Table 1”
Response 7: Thank you. “Table 2” was changed into“Table 1”. Please see line 84.
Remove from the Tables 1, 2, 3 and from Figures 1, 2, 3, 4, 5, 6 the section “CN, piglets fed a basal diet without any supplementation; CL-L, piglets fed the basal diet supplemented with 100 mg/kg coated lysozyme; CL-M, piglets fed the basal diet supplemented with 150 mg/kg coated lysozyme; CL-H, piglets fed the basal diet supplemented with 500 mg/kg coated lysozyme; UL, piglets fed the basal diet supplemented with 150 mg/kg lysozyme; Abs, piglets fed the basal diet supplemented with 150 mg/kg guitaromycin” instead the remaining parts of the legenda are useful for the understanding of each Table/Figure. The section above reported in italics takes several lines and is the same reported also in the abstract. I suggest not to repeat nine times the exact sentence in the 1 legenda of Tables and Figures but to mention it only at the beginning of the section 2.1.
Response 8: Thank you. I have removed “CN, piglets fed a basal diet without any supplementation; CL-L, piglets fed the basal diet supplemented with 100 mg/kg coated lysozyme; CL-M, piglets fed the basal diet supplemented with 150 mg/kg coated lysozyme; CL-H, piglets fed the basal diet supplemented with 500 mg/kg coated lysozyme; UL, piglets fed the basal diet supplemented with 150 mg/kg lysozyme; Abs, piglets fed the basal diet supplemented with 150 mg/kg guitaromycin”.and mention it only at the beginning of the section 2.1.
Line 211. Add a reference for the Chaol Index and Shannon Index
Response 9: Thank you. There was no significant difference between the results of Chaol Index and Shannon Index. For the sake of brevity, only Shannon Index is listed. Re-described in Results. Please see line 186 and line 401.
Line 358. The asterisk (*) reference in the test/table 4 is missing
Response 10: Thank you. I have marked * in the table 4. Please see line 328..
A final comment on the executed experimental protocol: a blank with the supplementation with only palm oil (used for the production of enveloped lysozyme) was not carried out. A comment on this point should be done by the Authors.
Response 11: Thank you. This coating is mainly palm oil and still contains other substances, so no palm oil control group is designed.
Minor suggestions and comments:
The terms “enveloped” and “coated” are used independently to indicate the “modified lysozyme”: are the two terms equivalent? Could be utilized only one term?
Response 12: Thank you. Yes, I make changes in the article. “Enveloped” was changed into “coated” in whole manuscript. Another reviewer also mentioned this issue, I also marked the word in blue.
As general recommendation: when an acronym is utilized the first time has to be explained
Response 13: Thank you. I have explained the abbreviations in section 2.1, please see lines 78-83.
Reviewer 2 Report
General comments
Lines 45-48: indeed weaning stress is probably the major predisposing factor for all these issues, however the pivotal role of certain bacteria species and in particular of E.Coli as a causative factor of the most symptoms postweaning. This is the reason why antibiotics were/are frequently used during this period. He authors need to revise this section.
Lines 51-52: the authors are not referring to the role of supplemented ZnO during this period, which is a common practice.
Lines 58-72: the authors do not provide any reference why they have chosen “guitaromycin” as a positive control treatment, and not another antibiotic e.g. colistin, amoxicillin.
Lines 285-298: the authors have determined the Antioxidant Indexes at a single sampling point, which however it is not specified in the materials and methods which one it is (day 1, 14 or 28?). Moreover, the effect on the antioxidant parameters is strogly correlated with the timepoint of sampling, as weaning stress occurs mostly around days 7-10 postweaning, and these parameters are affected by timepoint of sampling. How the authors can explain the effects they are reporting on the antioxidant parameters?
Line 254-onwards: there is not a specific reference for the explanation of the effects on short chain fatty acids and their significance.
Minor comments
Table 3: could have added a cumulative sum of all short chain fatty acids detected to see if the sum is different between treatments.
Author Response
Dear Reviewer,
Thank you very much for reviewing our manuscript (Antibiotics-1946742) and for your valuable comments and suggestions. They are very helpful for improving our manuscript. According to your comments and suggestions, we thoroughly revised our manuscript and submitted it to Antibiotics for further review. The changes based on your suggestions were highlighted in blue colour in the new version. Point by point corrections were made as follows:
Lines 45-48: indeed weaning stress is probably the major predisposing factor for all these issues, however the pivotal role of certain bacteria species and in particular of E.Coli as a causative factor of the most symptoms postweaning. This is the reason why antibiotics were/are frequently used during this period. He authors need to revise this section.
Response 1: Thank you very much for your valuable comments. I have changed this sentence to “Studies have shown that weaning stress can cause changes in the intestinal structure and function of piglets, as well as the intestinal flora, resulting in reduced immunity and oxidative stress, these factors would induce bacterial infections, especially E. coli infections, which ultimately leads to diarrhea, decreased growth performance, and, in severe cases, death.” Please see lines 47-48.
Lines 51-52: the authors are not referring to the role of supplemented ZnO during this period, which is a common practice.
Response 2: Thank you. I added ZnO to this sentence: “A large number of studies have confirmed that new feed additives, such as probiotics, enzymes, plant extracts, ZnO, and antibacterial agents, can effectively reduce the use of or replace feed antibiotics”. Please see lines 53-55.
Lines 58-72: the authors do not provide any reference why they have chosen “guitaromycin” as a positive control treatment, and not another antibiotic e.g. colistin, amoxicillin.
Response 3: Thank you. A large number of literatures show that guitarmycin as a feed additive can improve animal growth performance and prevent bacterial infection. It is included in the background of the world’s anti-antibiotic prohibition. Guitaromycin is also one of the most commonly used antibiotic additives. I list 2 similar literatures using guitarmycin as a positive control.
Ma J, Piao X, Shang Q, Long S, Liu S, Mahfuz S. Mixed organic acids as an alternative to antibiotics improve serum biochemical parameters and intestinal health of weaned piglets. Anim Nutr. 2021 Sep;7(3):737-749.
Ma J, Shang Q, Long S, Liu S, Mahfuz SU, Piao X. PSVII-10 Growth performance, serum biochemical parameters and intestinal health of piglets as affected by dietary moa at different levels. J Anim Sci. 2021 Oct 8;99(Suppl 3):410.
Lines 285-298: the authors have determined the Antioxidant Indexes at a single sampling point, which however it is not specified in the materials and methods which one it is (day 1, 14 or 28?). Moreover, the effect on the antioxidant parameters is strogly correlated with the timepoint of sampling, as weaning stress occurs mostly around days 7-10 postweaning, and these parameters are affected by timepoint of sampling. How the authors can explain the effects they are reporting on the antioxidant parameters?
Response 4: Thank you. “Day 28 sample...” was added in the manuscript. The main purpose of the article was to observe the effect of adding enveloped lysozyme on the antioxidant activity of piglets. And 7-10 days after weaning, the experiment was just beginning. Please see line 365.
Line 254-onwards: there is not a specific reference for the explanation of the effects on short chain fatty acids and their significance.
Response 5: Thank you. Added reference to valeric acid in the Discussion of the manuscript. I have changed "increase the short-chain fatty acid levels" into “increase the valeric acid levels in the cecum” in the article. Please see lines 282-283 and 420.
Reviewer 3 Report
The article is written in correct English. The experimental design is satisfactory and many blood, digestive and microbial parameters have been evaluated. It is a serious study relatively well presented.
Line 20 to 22 : dont repeat "fed a" or "fed the" I suggest "CN, basal diet....CH basal diet supplemented with 100,....."
Line 25 : "...globulin (Glob) plasma levels" instead of globulin (Glob) levels
Line 49 : "The accompanying problem" english to be improved
Line 55 : reference needed relative to the safety of Lyzozyme
Line 85 Body weights should be presented in kg instead of g to improve clarity. Values are given + or - STD wich is not necessary as soon as the global SEM is provided. Skip the +/- STD.
Line 85 any explanation of the strong decrease of feed intake for the UL group.
Line 86-94 These elements are associated with Table 1. Use a different font for Table 1 to distinguish from the main body text (as you did for table 3).
Line 149-157 same comment for table 2
Line 202 : Please provide the total VFA estimation. In total it seems that there was a global VFA reduction at least for acetic, propionic and butyric acids.
Line 274 : Give references supporting the fact that Plasma TP and globulines are linked with synthesis and metabolism of protein.
Line 283 : "sick" instead of "ill"
Line 311-315 This paragraph must be reviewed, because the general effect of Lysozyme in this study is a significant decrease of acetic, propionic and butyric acids and total VFAs...valeric also is only a slight part of total VFAS
Line 338 : What about the non coated product concentration in U/g ? Any way to assess the number of U/g really present in the finished feed ?
Line 435 : "coated" instead "envelopped"
Line 439 : "increase the short-chain fatty acid levels" according to above comments this is not true and must be skipped in the conclusion.
Author Response
Dear Reviewer,
Thank you very much for reviewing our manuscript (Antibiotics-1946742) and for your valuable comments and suggestions. They are very helpful for improving our manuscript. According to your comments and suggestions, we thoroughly revised our manuscript and submitted it to Antibiotics for further review. The changes based on your suggestions were highlighted in green colour in the new version. Point by point corrections were made as follows:
The article is written in correct English. The experimental design is satisfactory and many blood, digestive and microbial parameters have been evaluated. It is a serious study relatively well presented.
Line 20 to 22 : dont repeat "fed a" or "fed the" I suggest "CN, basal diet....CH basal diet supplemented with 100,....."
Response 1: Thank you very much for your valuable comments. “fed and fed the” in the abstract have been removed. Please see lines 20-22.
Line 25 : "...globulin (Glob) plasma levels" instead of globulin (Glob) levels
Response 2: Thank you. “globulin (Glob) levels” was changed into “globulin (Glob) plasma levels”. Please see line 25.
Line 49 : "The accompanying problem" english to be improved
Response 3: Thank you. I have changed some sentences. Please see lines 50-51.
Line 55 : reference needed relative to the safety of Lyzozyme
Response 4: Thank you. The safety of lysozyme as a dietary supplement in swine was evaluated by Schliffka et al. Reference 21. Please see lines 62-64.
Line 85 Body weights should be presented in kg instead of g to improve clarity. Values are given + or - STD wich is not necessary as soon as the global SEM is provided. Skip the +/- STD.
Response 5: Thank you. Sorry for the mistake, “g” was changed into “kg” in table 1, and supplement what is missing. and supplement what is missing. Please see line 92.
Line 85 any explanation of the strong decrease of feed intake for the UL group.
Response 6: Thank you. This result may be due to the poor palatability of lysozyme. Please see line 228.
Line 86-94 These elements are associated with Table 1. Use a different font for Table 1 to distinguish from the main body text (as you did for table 3)
Response 7: Thank you. The font for Table 1 was changed. Please see line 92.
Line 149-157 same comment for table 2
Response 8: Thank you. The font for Table 2 was changed. Please see line 139.
Line 202 : Please provide the total VFA estimation. In total it seems that there was a global VFA reduction at least for acetic, propionic and butyric acids.
Response 9: Thank you. The gas chromatograph used in our laboratory to measure VFA cannot measure total VFA. I have added a discussion of valeric acid and made changes to the conclusion. Please see line 282 and line 420.
Line 274 : Give references supporting the fact that Plasma TP and globulines are linked with synthesis and metabolism of protein.
Response 10: Thank you.The sentences that about “Plasma TP and globulines are linked with synthesis and metabolism of protein” were added in the manuscript. Please see lines 235-239.
Line 283 : "sick" instead of "ill"
Response 11: Thank you. I have changed ill to sick in the manuscript.. Please see line 250.
Line 311-315 This paragraph must be reviewed, because the general effect of Lysozyme in this study is a significant decrease of acetic, propionic and butyric acids and total VFAs.valeric also is only a slight part of total VFAS
Response 12: Thank you very much for your valuable comments. The gas chromatograph used in our laboratory to measure VFA cannot measure total VFA. And I have changed "increase the short-chain fatty acid levels" to increase the valeric acid levels in the cecum in the article. I have added a discussion of valeric acid and made changes to the conclusion. Please see line 282 and 420.
Line 338 : What about the non coated product concentration in U/g ? Any way to assess the number of U/g really present in the finished feed ?
Response 13: Thank you. Uncoated Product Vitality is 2×107 U/g. Please see lines 307-308. The method of detecting enzyme activity is Micrococcus luteus test.
Line 435 : "coated" instead "envelopped"
Response 14: Thank you. I have changed all the enveloped in the article to coated.
Line 439 : "increase the short-chain fatty acid levels" according to above comments this is not true and must be skipped in the conclusion.
Response 15: Thank you. I have changed "increase the short-chain fatty acid levels" to increase the valeric acid levels in the cecum in the manuscript. I have added a discussion of valeric acid and made changes to the conclusion. Please see line 282 and 420.
Round 2
Reviewer 1 Report
Dear Sirs
Enclosed you will find my evaluation of the revised manuscript “Effects of dietary enveloped lysozyme on the growth performance, antioxidant activity, immunity and gut health of weaned 3 piglets” of Xiangfei Xu et al.
It is my opinion that this article is suitable for the publication on Antibiotics after a minor revision at the following points:
· Line 62: is missing a sentence about the reference [22].
· Lines 62-64: please remove the sentence from “Experiments” to “lysozyme” in which is reported again the wrong sentence that Ferraboschi et al. carried out experiments on “antibiotic replacement by lysozyme”. The cited Authors (reference [21]) wrote a review on this topic without experimental activity.
Author Response
Dear Reviewer,
Thank you very much for reviewing our manuscript (Antibiotics-1946742) and for your valuable comments and suggestions. They are very helpful for improving our manuscript. According to your comments and suggestions, we thoroughly revised our manuscript and submitted it to Antibiotics for further review. The changes based on your suggestions were highlighted in red colour in the new version. Point by point corrections were made as follows:
- Line 62: is missing a sentence about the reference [22].
Response: Sorry for the mistake, and revision has been made.
- Lines 62-64: please remove the sentence from “Experiments” to “lysozyme” in which is reported again the wrong sentence that Ferraboschi et al. carried out experiments on “antibiotic replacement by lysozyme”. The cited Authors (reference [21]) wrote a review on this topic without experimental activity.
Response: Thanks for your suggestion, and revision has been made. I have reworked this sentence to “The safety of lysozyme as a dietary supplement in swine was evaluated by Schliffka et al.[21], who found no adverse effects, whereas it improved growth performance.” in the manuscript. Please see lines 61-63.
Reviewer 2 Report
The authors have revised the manuscript, although not as extensively required. Nevertheless, the current version is better than the initial one.
Author Response
The authors have revised the manuscript, although not as extensively required. Nevertheless, the current version is better than the initial one.
Response: Thanks for reviewing our manuscript (Antibiotics-1946742) and for your valuable comments and suggestions. They are very helpful for improving our manuscript. In the future, our research will be more in-depth and extensive.